# Nanoparticles, an Emerging Control Method for Harmful Algal Blooms: Current Technologies, Challenges, and Perspectives

**DOI:** 10.3390/nano13162384

**Published:** 2023-08-21

**Authors:** Jun Song, Zhibin Xu, Yu Chen, Jiaqing Guo

**Affiliations:** State Key Laboratory of Radio Frequency Heterogeneous Integration (Shenzhen University), College of Physics and Optoelectronic Engineering, Key Laboratory of Optoelectronic Devices and Systems of Ministry of Education and Guangdong Province, Shenzhen University, Shenzhen 518060, China; songjun@szu.edu.cn (J.S.); misakasagiri@gmail.com (Z.X.)

**Keywords:** nanoparticles, harmful algal bloom, photocatalysis, flocculation, water resource protection, water remediation

## Abstract

Harmful algal blooms (HABs) are a global concern because they harm aquatic ecosystems and pose a risk to human health. Various physical, chemical, and biological approaches have been explored to control HABs. However, these methods have limitations in terms of cost, environmental impact, and effectiveness, particularly for large water bodies. Recently, the use of nanoparticles has emerged as a promising strategy for controlling HABs. Briefly, nanoparticles can act as anti-algae agents via several mechanisms, including photocatalysis, flocculation, oxidation, adsorption, and nutrient recovery. Compared with traditional methods, nanoparticle-based approaches offer advantages in terms of environmental friendliness, effectiveness, and specificity. However, the challenges and risks associated with nanoparticles, such as their toxicity and ecological impact, must be considered. In this review, we summarize recent research progress concerning the use of nanoparticles to control HABs, compare the advantages and disadvantages of different types of nanoparticles, discuss the factors influencing their effectiveness and environmental impact, and suggest future directions for research and development in this field. Additionally, we explore the causes of algal blooms, their harmful effects, and various treatment methods, including restricting eutrophication, biological control, and disrupting living conditions. The potential of photocatalysis for generating reactive oxygen species and nutrient control methods using nanomaterials are also discussed in detail. Moreover, the application of flocculants/coagulants for algal removal is highlighted, along with the challenges and potential solutions associated with their use. This comprehensive overview aims to contribute to the development of efficient and sustainable strategies for controlling HAB control.

## 1. Introduction

Harmful algal blooms (HABs) involve the rapid growth of harmful algae on water surfaces and are mainly caused by water eutrophication and climate change [1,2]. Crucially, HABs reduce light availability to aquatic biota, resulting in lower oxygen levels in water bodies [3]. In addition, the toxins produced by algae damage the aquatic biota and reduce drinking water safety. A causative factor is the large amounts of wastewater containing nitrogen, phosphorus, and other nutrients that have been discharged into the natural environment with the development of industry and agriculture. In recent years, reports of HABs have become more frequent worldwide; for example, in Lake Taihu (China), Lake Winnipeg (Canada), and Lake Erie (USA), the frequent occurrence of HABs is threatening the ecological balance of these important water resources [4]. HABs significantly impact aquaculture, tourism, and public safety, and hence, appropriate measures are required to control them.

Studies have attempted to address the problem of HABs using different technologies, which are classified based on their characteristics. Traditional methods can be classified into physical, chemical, and biological (Figure 1). Physical methods are widely used in small water bodies. For example, mechanical pumps can be used to accelerate mixing between different water levels to break up and prevent the aggregation of algal cells [5]. In addition, cationic flocculants can be used to agglomerate algae and improve their sedimentation coefficients to transfer them to deep-water areas [6]. Other control methods include ultrasonic treatment, ultraviolet radiation, and membrane filtration [7,8,9]. However, owing to their high cost, these physical methods are only suitable for use in small water bodies and cannot be used in large sites, such as marine lakes. Chemical methods involve the use of algaecides or algal inhibitors such as CuSO_4_, Diuron, and H_2_O_2_ to control HABs [10] and are widely used because of their low cost and rapid effects. However, the addition of chemical agents can pollute the water environment; moreover, the destruction of the algal cell membranes by algaecides causes the release of toxins, thus reducing the environmental friendliness and sustainability of these methods. Biological methods involve the introduction of aquatic animals and microorganisms that compete with or prey on harmful algae into the water to rebalance the water environment [11]. However, these methods are relatively slow in terms of effectiveness and may cause species invasion in some situations [12,13,14]. Further, in experiments that released cladocerans and copepod zooplankton to control HABs, the results were not ideal [15].

Because of the rapid development of nanotechnology, nanoparticles have attracted considerable attention in the field of HAB control to kill or inhibit the algae causing HABs through various mechanisms, such as photocatalysis, flocculation and sedimentation, oxidation, adsorption, and nutrient recovery [16,17,18,19]. Compared with the aforementioned traditional anti-algae methods, these nano-based methods are more environmentally friendly, effective, and specific and, thus, show considerable promise as new anti-algae strategies. However, there are still some challenges and risks associated with using nanoparticles, such as their toxicity, bioaccumulation, and ecological impacts [20,21,22,23,24]. The aim of this review is to summarize the research progress on the use of nanoparticles to control HABs, compare the advantages and disadvantages of different types of nanoparticles, discuss the factors that influence their effectiveness and environmental impact, and suggest future directions for the research and development of this field.

## 2. Algae Blooms: Causes and Effects

### 2.1. Causes of Algae Blooms

Algal blooms have become more frequent in recent years as a result of human and natural factors [25]. Human activities, such as the leakage of agricultural fertilizers, industrial discharge, and urban sewage, are the main causes of water body eutrophication, and nutrients, such as NH_4_, NO_3_, and PO_4_, present in industrial wastewater are significant drivers of algal blooms [26,27]. Moreover, the introduction of invasive species into aquatic ecosystems can lead to extensive blooms in areas with reduced competition [28]. For example, HABs are often observed in ballast water discharge from ships and in the establishment of aquariums [29]. Further, climate change, particularly rising global temperatures, provides favorable conditions for the rapid growth of algae; algae thrive best at temperatures between 25 and 30 °C [30]. In addition, geological events, such as earthquakes and volcanic eruptions, also contribute to water body eutrophication by introducing large amounts of minerals, which encourage algal blooms [31].

### 2.2. Harmful Effects

The explosive growth of algae causes the coverage of a significant area of the water surface, depriving other aquatic biota of sunlight, resulting in decreased oxygen levels and the death of numerous aquatic organisms [32]. Moreover, the algae release various toxins that are consumed by shellfish such as oysters and are subsequently transferred through the food chain to fish, seabirds, and other organisms, including humans [33]. Specifically, these toxins can accumulate in humans through the consumption of contaminated seafood, leading to algal poisoning. Such poisoning incidents have been reported in several countries, including the United States of America, France, and Australia; each year, nearly 60,000 cases of poisoning resulting from shellfish are reported, including amnesic and paralytic shellfish poisoning, which are caused by domoic acid and saxitoxin, respectively. In addition to toxins, the metabolic algal byproducts are often foul smelling and impact the safety of drinking water [34]. Therefore, algal blooms pose a significant threat to public health and safety. Furthermore, the discoloration of water bodies caused by HAB, such as red tides in oceans and green algal blooms in lakes, severely affects the tourism and recreational industries [35]. In addition, the deterioration of water quality resulting from HAB significantly affects aquaculture, particularly oyster farming [36]. In 2011, oyster farms in Texas incurred losses of USD 10.3 million as a result of HAB.

## 3. Treatment Methods

The conditions required for the outbreak of HABs are an abundance of nutrients, favorable position within the food web, and suitable environmental conditions. Therefore, HABs can be prevented by disrupting these factors. After the occurrence of algal blooms, control measures involve direct salvaging, flocculation, and the destruction of growth conditions (Figure 1). Because of the increased frequency and severity of HAB in recent years, which have caused significant damage to the natural environment and human society, more attention should be paid to the prevention and control of these blooms [37,38,39].

### 3.1. Restricting Eutrophication

The algae responsible for HABs require a large quantity of nutrients in the aquatic environment [40]. In particular, NH_4_^+^, NO_3_^−^, and PO_4_^−^ are key nutrients and are found in industrial wastewater [41]. Despite being relatively expensive, microfiltration and ultrafiltration technologies are suitable for use in the wastewater outlets of smaller-scale factories [42,43]. Additionally, performing wastewater treatment before discharge can significantly reduce the occurrence of algal blooms near the discharge outlets [44]. In addition, the method of extracting bottom water using mechanical pumps and hydraulic mixers and flushing it at the water surface can achieve mixing at different water levels, ensuring the uniform dispersion of nutrients throughout the water body without allowing high concentrations to accumulate in specific areas [45]. However, this method not only consumes a significant amount of electricity but also poses the risk of pump clogging and corrosion. Ion exchange and ion adsorption technologies involve the introduction of adsorbents into a water body to recover phosphate ions selectively. This not only reduces the level of eutrophication but also suppresses algal growth. Further, the recovery of phosphorus helps alleviate its global shortage.

### 3.2. Biological Control

HABs are closely related to the competitive advantage of algae in the environment. Therefore, utilizing other organisms that can limit algal growth to mitigate the risk of HABs has great potential [46,47]. For example, the control of algal blooms using predators such as ciliates and flagellates has shown some effectiveness, and some algicidal bacteria and viruses have been reported to mitigate HABs [48]. Overall, the greatest advantage of biological control methods is their environmentally friendly nature [49]. However, the mechanisms of biological control are complex, and there is a lack of sufficient research in this area [50]. Importantly, their improper use may disrupt the ecological balance, and biological control takes time, making it more suitable as a preventive measure than post-treatment.

### 3.3. Disrupting Living Conditions

Chemical algicides such as copper sulfate, Diuron, and H_2_O_2_ have the advantage of rapid effectiveness against HABs [10]. However, their potential environmental impacts and safety concerns remain a subject of debate [51]. Further, unlike inland freshwater environments, marine environments are influenced by climate and ocean currents, making the use of chemical algicides challenging and risky. However, some oxidizing algicides such as CaO and H_2_O_2_ have advantages over others because they can eliminate the toxins released by harmful algae. In contrast, photocatalysis using TiO_2_ semiconductors to decompose water, produce radical species, and induce oxidative stress in algae has attracted attention as an environmentally friendly method for algal control [52]. Zhou et al. confirmed the inhibitory effect of ROS on *Microcystis aeruginosa*, a harmful alga, without harming other aquatic organisms [52]. However, as mentioned earlier, killing algal cells with algicides can cause cell lysis and the release of algal toxins, further reducing water quality. In comparison, the harvesting and coagulation–sedimentation methods are relatively mild and safe. In particular, artificial harvesting can quickly and safely remove algal blooms; however, it is generally only used in small-scale water bodies because of its high cost [5,53]. Currently, coagulation–sedimentation is recognized as one of the most promising methods for combating harmful algal blooms [54,55]. In this method, the negatively charged algal cells are aggregated into flocs through electrostatic adsorption or charge attraction, causing the cells to sink to the bottom of the water [56]. The use of biocoagulants, such as chitosan, modified clays, and cationic coagulants, has also been reported to have significant effects on the control of HABs, and the combination of multiple coagulants enhances the potential of this method [57,58]. Certain secondary metabolites found in plants, including glycosides, polyphenols, and polysaccharides, have been reported to improve the adsorption of clay by algae.

## 4. Nanoparticles to Control HABs

### 4.1. Photocatalysis to Produce ROS

Photocatalysis utilizing TiO_2_ semiconductor electrodes to decompose water under light irradiation and generate free radicals has gained significant attention as a promising approach to combat algal proliferation [59]. Upon exposure to photons, the electrons in the photocatalytic semiconductors undergo excitation and transition from the valence band to the conduction band, thereby creating electron vacancies (holes) within the valence band (Figure 2) that can react with other compounds to produce radical species. To enhance the efficiency and sustainability of photocatalysis, researchers have explored strategies such as doping TiO_2_ semiconductors with nitrogen and phosphorus [60], as reported by Wang et al. As such, they could utilize abundant natural light effectively while maintaining the environmental sustainability of the TiO_2_ photocatalyst [61]. Crucially, nanoparticles offer distinct advantages over traditional photocatalytic semiconductors because of their large specific surface areas [62]. Further, the ability to design nanomaterials precisely enables the adjustment of their bandgap, energy levels, and surface activity, thereby improving their photocatalytic efficiency and ability to generate reactive oxygen species (ROS). These unique features give nanomaterial-based photocatalysis significant advantages over conventional methods employed for algal removal.

As a representative photocatalytic semiconductor, TiO_2_ nanoparticles also have strong anti-algae capabilities. For example, a TiO_2_ nanophotocatalyst prepared by Pinho et al., destroyed *M. aeruginosa* but also broke down intracellular and extracellular microcystins [63]. Jin et al. combined photocatalysis and flocculation methods and added N-TiO_2_ to a flocculant to enable the self-purification of algae flocs under visible-light irradiation. After application, the *M. aeruginosa* completely precipitated in 10 min, and 97% of the algae cells were killed after 32 h of visible-light irradiation [64]. However, poor transparency and difficulty in recycling the powdered nanophotocatalytic enzymes limit their practical application. Fan et al. loaded Ag_2_MoO_4_/TACN nanoparticles onto loofah using an oscillating impregnation method to form a floating photocatalyst that achieved 100% chlorophyll removal within 4 h of illumination [65]. However, placing metal-based nanoparticles in water bodies poses the risk of environmental pollution; therefore, non-metallic-based nanoparticles are often used as green photocatalysts. As such, graphitic carbon nitride (g-C_3_N_4_) nanomaterials having visible-light responsiveness, chemical stability, and high specific surface area are ideal anti-algae material. For example, Song et al. prepared a floating photocatalyst by depositing g-C_3_N_4_ on expanded perlite using facile implantation catalysis. Notably, this method prevented the secondary pollution caused by microcystins [66]. Further, the use of composite magnetic nanoparticles can effectively improve the recyclability of photocatalysts, reduce pollution, and reduce costs. Qi et al. synthesized a recyclable magnetic Zn-doped Fe_3_O_4_ visible-light-catalytic enzyme to achieve algal killing and recovery. At a catalyst dosage of 0.05 g/L, the algal removal efficiency of the catalyst after three regeneration cycles only slightly decreased [67]. The construction of heterostructures can improve the separation of photogenerated electron–hole pairs and is considered an effective method for improving the ROS yield. Of the available materials for preparing heterostructures, Ag_3_PO_4_ produces a very high quantum yield of up to 90% at wavelengths longer than 420 nm, giving it considerable potential. For example, He et al. proposed that a Z-type heterostructure formed between Ag_3_PO_4_ and g-C_3_N_4_ could promote the separation of electron–hole pairs [68]. Furthermore, by taking advantage of the strong paramagnetism of ZnFe_2_O_4_, as well as its narrow bandgap, low toxicity, photocatalytic stability, and other advantages, Fan et al. successfully prepared ternary nanocomposites of ZnFe_2_O_4_/Ag_3_PO_4_/g-C_3_N_4_ that were also magnetic, enabling their separation, using an in situ chemical precipitation method. Crucially, the prepared composite combined the advantages of these nanomaterials. Therefore, composite methods are promising for improving algal removal [69].

Metal–organic frameworks (MOFs) are highly versatile materials composed of inorganic nodes interconnected by organic linkers that form a porous crystalline structure. This unique architecture enables the precise tuning of MOF properties by modifying their composition and structure. As a result, MOFs have gained significant attention as an anti-algae technology owing to their exceptional characteristics and potential. Some of the key challenges in photocatalysis are the limited utilization of sunlight, inadequate exposure of active sites, and lack of control over catalytic processes. MOFs address these issues because of their high photon capture efficiency, large specific surface area, and adjustable porosity, making them highly effective photocatalytic nanomaterials. Fan et al. utilized an MOF framework to achieve a high inactivation rate of 93.1% at a low dose of 10 mg/L and demonstrated that superoxide radicals were the dominant active species for algal inactivation, although other ROS were also involved [70]. Wang et al. prepared g-C_3_N_4_/Cu-MOF nanocomposites and studied the effect of g-C_3_N_4_ doping ratio on the photocatalytic efficiency of the Cu-MOF framework. They found that moderate doping (10 wt%) improved the photocatalytic efficiency, whereas excessive doping (20 wt%) suppressed the generation of free radicals owing to the high electron–hole recombination rate [71]. In 2022, Hu et al. reported that SNP-TiO_2_ nanoparticles had an inhibitory effect on the growth of harmful algae in *Karenia mikimotoi* by 81.8%. The MOF architecture was further used to reduce the band gap to 2.82 eV, improving the photocatalytic performance SNP-TiO2@Cu-MOF. As a result, the inactivation efficiency toward *K. mikimotoi* reached 93.75% [72].

Overall, the utilization of photocatalysis for ROS production presents a promising avenue for the development of efficient and sustainable solutions for treating HABs. Table 1 shows a comparison of the different photocatalysts used for algal control.

### 4.2. Nutrient Control Methods

Algal blooms are often caused by eutrophication, which occurs due to the discharge of industrial and agricultural sewage into water bodies [81]. One mitigating strategy, nutrient ion recovery is particularly intriguing; it combines recycling (that is, the recovery of phosphorous) with a reduction in the key nutrient that drives HABs [29].

Ion exchange methods are one class of methods to achieve phosphorus recovery, but they are often limited by the high cost and selectivity of the media used [82]. The development of nanomaterials has overcome these challenges. Recent studies have highlighted the use of iron oxide nanoparticles in ion exchange technology (HIX) as an efficient and responsive approach to phosphorus removal [83].

In addition, adsorption is another commonly employed method for phosphorus recovery, involving adsorption and subsequent ion transfer [84]. For example, phosphorus is initially adsorbed onto the surface of the adsorbent and then transported within the adsorbent material. An ideal phosphorus removal adsorbent should have several key characteristics, including a large specific surface area, high adsorption capacity, suitable pore size for adsorption, stability, biocompatibility, environmental friendliness, and ease of operation [85]. Nanoadsorbents, in addition to meeting these conditions, can provide binding sites for phosphorus through surface modification, thereby improving the selectivity and adsorption activity [44]. Different types of phosphorus removal nanoparticles exhibit unique properties [86]. For instance, ZnO particles are soluble, enabling the release of Zn ions capable of inhibiting algal growth [87].

Some new basic metal nanoparticles, including lanthanum and nano-zero-valent iron (nZVI), have been found to have good absorption ability. Due to its relatively low price and abundant reserves among rare-earth elements, lanthanum is considered a promising rare earth element [88]. Further, owing to its special affinity for phosphate and its efficient phosphorus removal, it has received widespread attention for phosphorus recovery. Zhang et al. synthesized a lanthanum hydroxide nanoadsorbent with high phosphorus adsorption efficiency and studied the mechanism of phosphate adsorption via a macro experiment. Ligand exchange, surface co-precipitation, electrostatic interactions, and Lewis acid–base interactions were found to be the main mechanisms of phosphate adsorption, and these are strongly related to the solution pH [89]. The recovery and separation of lanthanum nanoparticles can be improved by loading them with magnetic materials. For example, Chen et al. reported that magnetic mesoporous silica nanoparticles (Mag MSNs) with 42% La, synthesized by loading La(OH)_3_ into silica nanoparticles, exhibited ultrahigh stability in the pH range of 4–11 [90]. Additionally, due to its magnetic core, it is easier to recycle and separate, and its working principle is shown in Figure 3. nZVI has received widespread attention as an adsorbent owing to its low cost, nontoxicity, and high specific surface area.

Nanoadsorbents based on nZVI remove phosphate through mechanisms such as electrostatic attraction, ion exchange, and chemical adsorption, with high removal rates. Maamoun et al. studied the adsorption kinetics of NZVI using adsorption isotherms and kinetic modeling, and proved that the phosphorus removal process of NZVI is mainly chemical adsorption, and the physical deposition on the surface can occur in two steps: liquid film and liquid–solid diffusion [91]. However, although nZVI nanoparticles have excellent phosphate removal efficiency, they often need to be stored in methanol, ethanol, or other anaerobic environments owing to their strong reducibility. Zhou et al. prepared nZVI/SCB superparamagnetic composites using a liquid-phase reduction method that remained stable for at least 450 days in sealed bags [92].

Overall, combining nanomaterials is highly promising for efficient phosphorus removal and nutrient recycling, addressing the underlying causes of algal blooms, and contributing to the overall health and sustainability of water bodies. Table 2 shows a comparison of the performance of different phosphorous nanoadsorbents.

### 4.3. Flocculation/Coagulant-Based Algae Removal

Flocculants and coagulants are commonly used in the treatment of HABs. Traditional flocculants, such as alum, iron(III) chloride, and modified clay, can aggregate and sediment algal cells through electrostatic adsorption, thus temporarily improving water clarity [102,103]. However, the reliability of electrostatic adsorption is limited by the negative charge present on the surface of algal cells. In contrast, nanocationic flocculants offer a superior flocculation performance for water treatment. This is attributed to their larger specific surface area, higher positive charge density, and enhanced adsorption and bridging effects compared to those of traditional inorganic flocculants [104]. The use of aluminum- and iron-based flocculants has shown algal removal efficiencies ranging from 85% to 95% [105]. Nonetheless, in water bodies with low algal density, the flocculation density may not be sufficient to deposit algae completely, which can result in their resuspension [106]. To address this issue, Ma et al. made further advancements by employing a nanomagnetic composite flocculant Fe_3_O_4_/CPAM (ionic polyacrylamide) [107]. This technology enhances flocculation, and the magnetic properties facilitate recovery and recyclability, enabling the effective removal of algae after flocculation and settlement. The use of positively charged nanoparticles on the surface can also achieve better flocculation effects. The Pd/g-C_3_N_4_ nanoparticles developed by Lu et al. have opposite charges on the surface of algal cells, which can adsorb algal cells more tightly and cause damage to the cell membrane, leading to complete cell death [69]. The principle is shown in Figure 4.

Compared to traditional aluminum- and iron-based coagulants, natural coagulants offer advantages such as high efficiency, cost-effectiveness, biodegradability, and environmental friendliness, making them widely used in water treatment chemicals. Among the natural polymers, chitosan has garnered increasing attention as a promising natural polymer for coagulant preparation because of its low toxicity and excellent biodegradability. Chen et al. modified chitosan through photo polymerization and prepared a coagulant having excellent algal removal performance [108]. However, the deposited algal blooms cannot be removed from the water and also release algal toxins, endangering other aquatic organisms. Therefore, the use of settling flocculants requires a combination of filtration, flotation, and other methods to treat the algae [109]. Because of the reliance on complex and expensive equipment for filtration systems and the aeration devices required for flotation, the cost of using flocculent settling agents for treating toxic algae is relatively high [110].

Another potential research direction is to tackle algal toxins. HABs are often associated with harmful extracellular organic matter (EOM) that can be toxic to aquatic organisms. Using coagulation, Yang et al. achieved a high coagulation performance using “tanfloc” and simultaneously addressed the problem of EOM [111].

In addition, flotation technology can be used. Flotation technology was first used in the United Kingdom and involves the introduction of oil droplets into water. The undesired compounds adhere to the oil droplets, and, owing to the lower density of oil compared with water, they rise to the water surface. In addition, some nanomaterials can lower the density of flocs by generating small bubbles, causing them to float on the water surface. Lin et al. developed a novel self-supporting chitosan dual-functional nanoparticle, CaO_2_@PEG-loaded, that can rapidly flocculate algae and naturally float flocs by continuously releasing oxygen [112]. Li et al. conducted a study on the risk of algal toxin release within coagulation flocs. They synthesized an NH_2_-MIL-101(Cr) MOF that exhibited excellent stability and negligible risk of rapid toxin release. This paved the way for the application of MOF in controlling HABs [113].

In summary, the use of flocculants and coagulants for algal control has shown excellent results and requires very small dosages to remove large quantities of algae effectively. Compared with other methods, coagulation/flocculation is more cost-effective and environmentally friendly. Further, it is particularly suitable for applications in small water bodies owing to its relative safety. However, in large water bodies, the challenge lies in handling algal toxins released from the ruptured algal cells within flocs because capturing and removing flocs becomes more difficult. Currently, mainstream approaches include toxin inactivation using oxidants, flotation separation, and pre-oxidation. Table 3 compares the performances of various flocculants/coagulants.

## 5. Conclusions and Outlook

In this review, we have summarized the recent advances in nanomaterials for HAB control, a research area that has grown significantly in recent years (Figure 5). We have examined the current state of the art from two perspectives: the anti-algae mechanisms and types of nanoparticles. Our literature review reveals that studies have explored various anti-algae methods based on nanoparticles, which can exploit photocatalytic properties to generate (ROS) that induce oxidative stress and inhibit algal growth. In particular, photocatalysis with nanomaterials, particularly TiO_2_, has shown potential in combating algal proliferation. Metal and metal oxide-based nanoparticles (NPs), such as AgNPs, CuNPs, TiO_2_ NPs, ZnO NPs, and iron oxide NPs, exhibit high toxicity toward algae cells and possess anti-algae and photocatalytic properties that produce ROS. Nanoparticles can also address the root causes of algal blooms by recovering nutrient ions, particularly phosphorus, through ion exchange and adsorption processes. These innovative approaches offer ecofriendly and sustainable solutions for HAB control. Moreover, compared to conventional inorganic flocculants, nanocationic flocculants can neutralize the negative charge on the algal cell surface, achieving better flocculation. By combination with magnetic nanoparticles, the further recovery of flocs and the reuse of flocculants is possible. Despite the benefits of nanoparticle-based approaches, it is crucial to evaluate the potential risks associated with their use, such as toxicity, bioaccumulation, and ecological impacts. Future research should focus on optimizing nanoparticle properties, discovering new nanomaterials, and developing techniques for enhanced algal inhibition and removal. Additionally, comprehensive assessments of the environmental risks and long-term effects of nanoparticle use are necessary. By addressing these challenges and pursuing sustainable and effective strategies, nanoparticles can revolutionize HAB control and contribute to the conservation of aquatic ecosystems. The integration of nanoparticles with other control methods, such as physical, chemical, and biological approaches, as well as the development of ecofriendly synthesis methods, will further improve their efficacy and sustainability. Through continued research and innovation, nanoparticles can play a significant role in mitigating the harmful impacts of algal blooms and ensuring the health and balance of aquatic environments.

## Figures and Tables

**Figure 1 nanomaterials-13-02384-f001:**
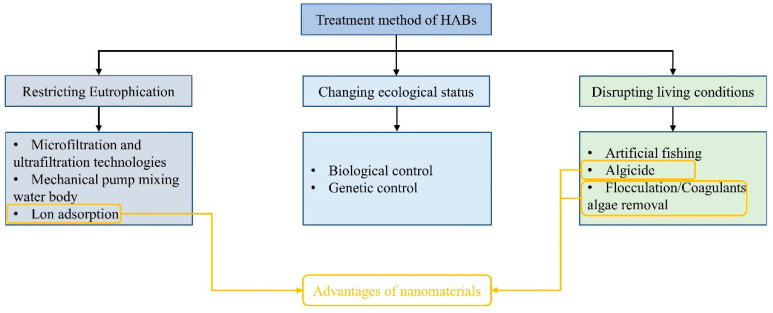
HAB treatment methods.

**Figure 2 nanomaterials-13-02384-f002:**
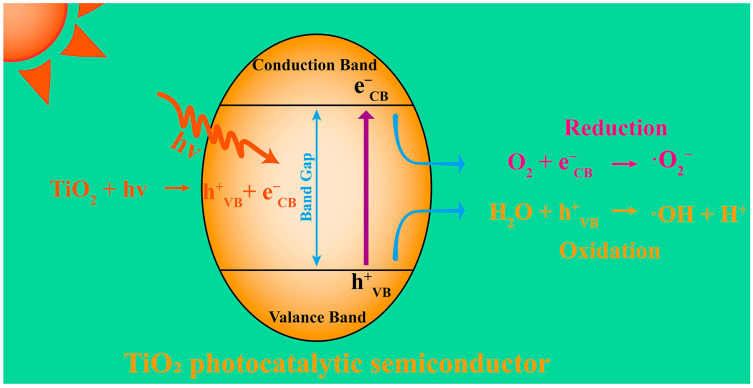
Principle of ROS generation by semiconductor photocatalysts.

**Figure 3 nanomaterials-13-02384-f003:**
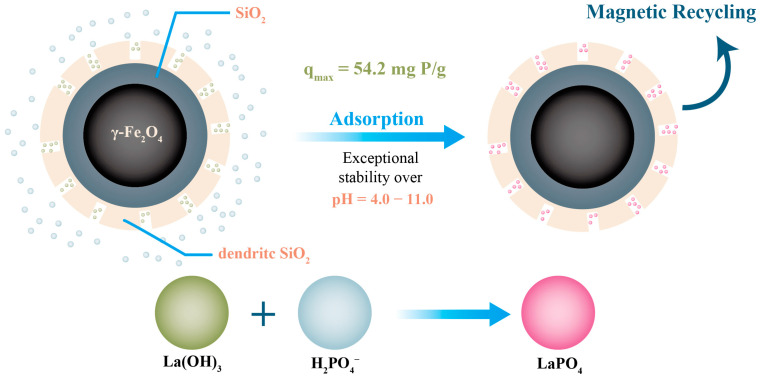
The process of phosphorus recovery from magnetic composite nanoparticles.

**Figure 4 nanomaterials-13-02384-f004:**
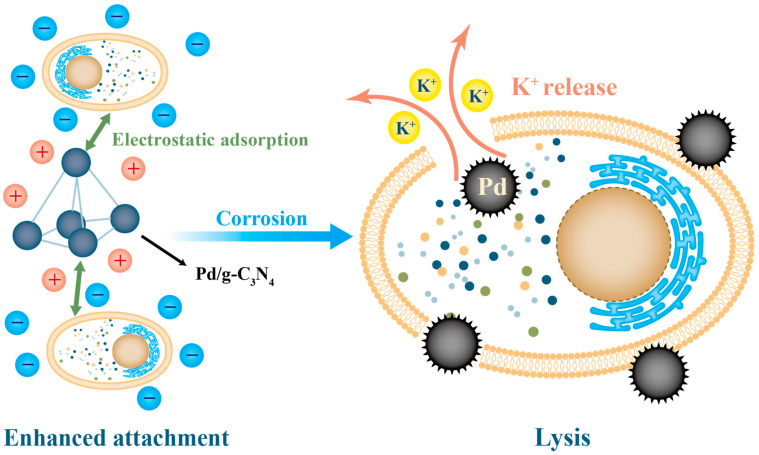
Proposed mechanism on catalytic inactivation of M. aeruginosa in the presence of Pd/g-C_3_N_4_.

**Figure 5 nanomaterials-13-02384-f005:**
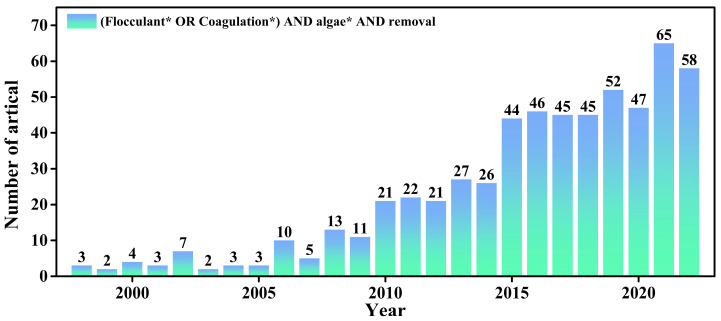
Numbers of scientific publications per year containing keywords “flocculant* OR coagulation*” and “algae*” and “removal*” from 1998 to 2022. (* Wildcard used for retrieval, representing different forms of the same word.)

**Table 1 nanomaterials-13-02384-t001:** Comparison of Different Photocatalysts.

Photocatalyst	Algae Specie	Algicidal Rate (%)	Action Time (h)	Dose (mg/L)	Reference
TiO_2_	*Microcystis aeruginosa*	-	-	100	Phinho et al. (2015) [63]
F-Ce-TiO_2_/EP450	*Chattonella marina*	98.10	9	4000	Wang et al. (2017) [73]
N-TiO_2_	*Microcystis aeruginosa*	97	32	200	Jin et al. (2018) [64]
g-C_3_N_4_	*Microcystis aeruginosa*	74.4	6	2000	Song et al. (2018) [66]
NP-TiO_2_/C	*Microcystis aeruginosa*	92.6	6	-	Wang et al. (2019) [74]
Fe_2_O_3_-TiO_2_	*Chlorella vulgaris*	99	24	25	Baniamerian et al. (2019) [75]
Ag/AgCl@ZIF-8	*Microcystis aeruginosa*	93.1	6	10	Fan et al. (2020) [70]
TiO_2_	*Microcystis aeruginosa*	-	-	-	Lee et al. (2020) [76]
Zn-doped Fe_3_O_4_	*Microcystis aeruginosa*	96	6	50	Qi et al. (2020) [67]
Cu_2_(OH)PO_4_	*Microcystis aeruginosa*	90.40	3	32	Asogdom et al. (2021) [77]
Ag/AgCl@C_4_N_4_@UIO-66(NH_2_~)	*Microcystis aeruginosa*	99.90	3	30	Fan et al. (2021) [78]
ZnFe_2_O_4_/Ag_3_PO_4_/g-C_3_N_4_	*Microcystis aeruginosa*	94.31	3	100	Fan et al. (2022) [69]
TiO_2_	*Alexandrium minutum*	75.1 ± 13.8	72	-	Ibrahim et al. (2022) [79]
Bi_2_O~3@CU-MOF	*Karenia mikimotoi*	96.35	4	60	Wang et al. (2022) [72]
Ag_2_MoO_4_/TACN@LF	*Microcystis aeruginosa*	100	4	6000	Fan et al. (2023) [65]
g-C_3_N_4_/Cu-MOF	*Microcystis aeruginosa*	92.4	6	6	Wang et al. (2023) [71]
TiO_2_/Ag_3_PO_4_	*Cylindrospermopsis raciborskii*	91.75	5	300	Zhou et al. (2023) [80]
SNP-TiO_2_@Cu-MOF	*Karenia mikimotoi*	93.75	6	100	Hu et al. (2023) [72]

**Table 2 nanomaterials-13-02384-t002:** Comparison of Different Adsorbents.

Adsorbent	Adsorption Capacity (mg/g)	Removal Rate	Action Time (min)	pH Value	Dose (mg/L)	Reference
MgO-D	73.8	161 mg/g	120	7	300	Xia et al. (2016) [93]
Porous MgO	-	236 mg/g	180	5	100	Ahmed et al. (2017) [94]
CSH@SiO2@MgO	-	93.9 mg/g	60	8	400	Si et al. (2017) [95]
LBR-Zr	72.8	65.8%	60	6	1250	Zong et al. (2018) [96]
Lanthanum-based flocculant						
Mag-MSNs-42%La	54.2	-	100	4–10	500	Chen et al. (2018) [90]
NCS@ZSM-5-H/La	-	144.92 mg/g	20	4	500	Salehi et al. (2020) [97]
La@201	-	122 mg/g	1440	4	500	Zhang et al. (2021) [89]
LC@ARE (1:2)	77.43	91 mg/g	720	7	500	Teea et al. (2022) [98]
nZVI-based flocculant						
Alginate-nZVI	-	60%	30	6.5	5000	Ahmed et al. (2018) [99]
Chitosan-coated nZVI	437	80%	30	5	300	Shanableh et al. (2019) [100]
RSBC-nZVI	12.14	-	180	3–8	2500	Ma et al. (2020) [101]
nZVI	-	76.8%	180	7	1000	Maamoun et al. (2020) [91]
Sugarcane bagasse nZVI	205.2	98.6%	90	3	1600	Zhou et al. (2022) [92]

**Table 3 nanomaterials-13-02384-t003:** Comparison of Different Flocculants/Coagulants.

Flocculants/ Coagulants	Algae Specie	Removal Rate	Action Time (min)	pH	Dose (mg/L)	Reference
CTA-DMDAAC	*Microcystis aeruginosa*	98.80	20	7	4	Chen et al. (2018) [108]
TiCl_4_	*Microcystis aeruginosa*	85.00	5	-	60	Xu et al. (2018) [105]
MHCS-g-P	Multiple algae species	93.60	15	7–8	4	Chen et al. (2019) [114]
Fe3O4/CPAM	*Chlamydomonas*	97.00	9	4–9	1.2	Ma et al. (2019) [107]
CaO2@PEG	Multiple algae species	98.84	120	10	8	Lin et al. (2021) [112]
SPC/Fe^2+^	*Microcystis aeruginosa*	98.50	5	-	56	Tian et al. (2021) [115]
AM-DMDAAC	*Microcystis aeruginosa*	90.00	10	3–11	4	Yang et al. (2021) [111]
Fe (VI)	*Microcystis aeruginosa*	92.60	20	-	0.8	Jin et al. (2022) [116]
NH2-MIL-101(Cr) MOFs	*Microcystis aeruginosa*	95.00	90	4–10	30	Li et al. (2022) [113]
Pd/g-C_3_N_4_	*Microcystis aeruginosa*	95.17	10	7	4000	Lu et al. (2022) [117]
CAFM	*Microcystis aeruginosa*	96.00	17	7.5–8.5	40	Ma et al. (2022) [118]
TCCs	*Microcystis aeruginosa*	90.00	15	7–10	4	You et al. (2022) [119]
PAD-g-MNC	Multiple algae species	97.31	30	4–11	5	Du et al. (2023) [120]

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
