# Peer review of "Nanoparticles, an Emerging Control Method for Harmful Algal Blooms: Current Technologies, Challenges, and Perspectives"

_nanomaterials, 2023, doi:10.3390/nano13162384_

Round 1
Reviewer 1 Report
The paper is interesting and can be published.
However, the number of rated papers is too small for a review article. I recommend you to complete the list of bibliographic citations with other works.
Author Response
Reviewer #1:
The paper is interesting and can be published.
However, the number of rated papers is too small for a review article. I recommend you to complete the list of bibliographic citations with other works.
Response:
Thank you very much for your kind and valuable suggestions. We have added numerous references to support the comprehensiveness of the content in the article.
Reference
[7] P. Rajasekhar, L. H. Fan, T. Nguyen, F. A. Roddick, A review of the use of sonication to control cyanobacterial blooms. Water Research 46, 14(2012) 4319-4329.
[8] M. L. Li, D. Y. Chen, Y. Liu, C. Y. Chuang, F. Z. Kong, P. J. Harrison, X. S. Zhu, Y. L. Jiang, Exposure of engineered nanoparticles to Alexandrium tamarense (Dinophyceae): Healthy impacts of nanoparticles via toxin-producing dinoflagellate. Sci. Total Environ. 610, (2018) 356-366.
[9] Y. H. Park, S. Kim, H. S. Kim, C. Park, Y. E. Choi, Adsorption Strategy for Removal of Harmful Cyanobacterial Species Microcystis aeruginosa Using Chitosan Fiber. Sustainability 12, 11(2020).
[12] R. Sun, P. F. Sun, J. H. Zhang, S. Esquivel-Elizondo, Y. H. Wu, Microorganisms-based methods for harmful algal blooms control: A review. Bioresource Technology 248, (2018) 12-20.
[13] H. Mohan, S. Vadivel, S. Rajendran, Removal of harmful algae in natural water by semiconductor photocatalysis- A critical review. Chemosphere 302, (2022).
[14] F. R. Chen, Z. G. Xiao, L. Yue, J. Wang, Y. Feng, X. S. Zhu, Z. Y. Wang, B. S. Xing, Algae response to engineered nanoparticles: current understanding, mechanisms and implications. Environmental Science-Nano 6, 4(2019) 1026-1042.
[19] V. Aruoja, S. Pokhrel, M. Sihtmae, M. Mortimer, L. Madler, A. Kahru, Toxicity of 12 metal-based nanoparticles to algae, bacteria and protozoa. Environmental Science-Nano 2, 6(2015) 630-644.
[20] Y. X. Wang, X. S. Zhu, Y. M. Lao, X. H. Lv, Y. Tao, B. M. Huang, J. X. Wang, J. Zhou, Z. H. Cai, TiO2 nanoparticles in the marine environment: Physical effects responsible for the toxicity on algae Phaeodactylum tricornutum. Sci. Total Environ. 565, (2016) 818-826.
[21] R. J. Miller, E. B. Muller, B. Cole, T. Martin, R. Nisbet, G. K. Bielmyer-Fraser, T. A. Jarvis, A. A. Keller, G. Cherr, H. S. Lenihan, Photosynthetic efficiency predicts toxic effects of metal nanomaterials in phytoplankton. Aquatic Toxicology 183, (2017) 85-93.
[22] E. Priyadarshini, S. S. Priyadarshini, N. Pradhan, Heavy metal resistance in algae and its application for metal nanoparticle synthesis. Applied Microbiology and Biotechnology 103, 8(2019) 3297-3316.
[24] A. H.M, M. N. A. Uda, S. C. B. Gopinath, Z. A. Arsat, F. Abdullah, M. F. A. Muttalib, M. K. R. Hashim, U. Hashim, M. N. A. Uda, A. R. W. Yaakub, N. H. Ibrahim, N. A. Parmin, T. Adam, Green route synthesis of antimicrobial nanoparticles using sewage alga bloom. Materials Today: Proceedings (2023).
[25] H. W. Paerl, T. G. Otten, Harmful Cyanobacterial Blooms: Causes, Consequences, and Controls. Microbial Ecology 65, 4(2013) 995-1010.
[26] G. Hallegraeff, H. Enevoldsen, A. Zingone, Global harmful algal bloom status reporting. Harmful Algae 102, (2021) 3.
[27] H. W. Paerl, J. Huisman, Climate change: a catalyst for global expansion of harmful cyanobacterial blooms. Environmental Microbiology Reports 1, 1(2009) 27-37.
[28] W. W. Carmichael, Health effects of toxin-producing cyanobacteria: "The CyanoHABs". Human and Ecological Risk Assessment 7, 5(2001) 1393-1407.
[29] M. Pal, P. J. Yesankar, A. Dwivedi, A. Qureshi, Biotic control of harmful algal blooms (HABs): A brief review. Journal of Environmental Management 268, (2020).
[30] T. W. Davis, D. L. Berry, G. L. Boyer, C. J. Gobler, The effects of temperature and nutrients on the growth and dynamics of toxic and non-toxic strains of Microcystis during cyanobacteria blooms. Harmful Algae 8, 5(2009) 715-725.
[31] D. G. Schmale, A. P. Ault, W. Saad, D. T. Scott, J. A. Westrick, Perspectives on Harmful Algal Blooms (HABs) and the Cyberbiosecurity of Freshwater Systems. Frontiers in Bioengineering and Biotechnology 7, (2019).
[32] Y. Z. Tang, H. F. Gu, Z. H. Wang, D. Y. Liu, Y. Wang, D. D. Lu, Z. X. Hu, Y. Y. Deng, L. X. Shang, Y. Z. Qi, Exploration of resting cysts (stages) and their relevance for possibly HABs-causing species in China. Harmful Algae 107, (2021).
[33] N. Xu, M. Wang, Y. Z. Tang, Q. Zhang, S. S. Duan, C. J. Gobler, Acute toxicity of the cosmopolitan bloom-forming dinoflagellate Akashiwo sanguinea to finfish, shellfish, and zooplankton. Aquatic Microbial Ecology 80, 3(2018) 209-222.
[34] N. Inaba, V. L. Trainer, Y. Onishi, K. I. Ishii, S. Wyllie-Echeverria, I. Imai, Algicidal and growth-inhibiting bacteria associated with seagrass and macroalgae beds in Puget Sound, WA, USA. Harmful Algae 62, (2017) 136-147.
[35] C. R. C. Kouakou, T. G. Poder, Economic impact of harmful algal blooms on human health: a systematic review. Journal of Water and Health 17, 4(2019) 499-516.
[36] C. Belin, D. Soudant, Z. Amzil, Three decades of data on phytoplankton and phycotoxins on the French coast: Lessons from REPHY and REPHYTOX. Harmful Algae 102, (2021).
[37] A. Zingone, L. Escalera, K. Aligizaki, M. Fernandez-Tejedor, A. Ismael, M. Montresor, P. Mozetic, S. Tas, C. Totti, Toxic marine microalgae and noxious blooms in the Mediterranean Sea: A contribution to the Global HAB Status Report. Harmful Algae 102, (2021).
[38] S. Sakamoto, W. A. Lim, D. D. Lu, X. F. Dai, T. Orlova, M. Iwataki, Harmful algal blooms and associated fisheries damage in East Asia: Current status and trends in China, Japan, Korea and Russia. Harmful Algae 102, (2021).
[39] C. Wang, Z. Y. Wang, P. F. Wang, S. H. Zhang, Multiple Effects of Environmental Factors on Algal Growth and Nutrient Thresholds for Harmful Algal Blooms: Application of Response Surface Methodology. Environmental Modeling & Assessment 21, 2(2016) 247-259.
[40] I. Sunesen, S. M. Mendez, J. E. Mancera-Pineda, M. Y. D. Bottein, H. Enevoldsen, The Latin America and Caribbean HAB status report based on OBIS and HAEDAT maps and databases. Harmful Algae 102, (2021).
[41] T. C. Daniel, A. N. Sharpley, J. L. Lemunyon, Agricultural phosphorus and eutrophication: A symposium overview. Journal of Environmental Quality 27, 2(1998) 251-257.
[42] M. Rathod, K. Mody, S. Basha, Efficient removal of phosphate from aqueous solutions by red seaweed, Kappaphycus alverezii. Journal of Cleaner Production 84, (2014) 484-493.
[43] B. L. Wu, J. Wan, Y. Y. Zhang, B. C. Pan, I. M. C. Lo, Selective Phosphate Removal from Water and Wastewater using Sorption: Process Fundamentals and Removal Mechanisms. Environ. Sci. Technol. 54, 1(2020) 50-66.
[44] X. Y. Li, Y. H. Xie, F. Jiang, B. Wang, Q. L. Hu, Y. Tang, T. Luo, T. Wu, Enhanced phosphate removal from aqueous solution using resourceable nano-CaO2/BC composite: Behaviors and mechanisms. Sci. Total Environ. 709, (2020).
[45] P. Loganathan, S. Vigneswaran, J. Kandasamy, N. S. Bolan, Removal and Recovery of Phosphate From Water Using Sorption. Critical Reviews in Environmental Science and Technology 44, 8(2014) 847-907.
[46] W. J. Xu, J. T. Wang, L. J. Tan, X. Guo, Q. N. Xue, Variation in allelopathy of extracellular compounds produced by Cylindrotheca closterium against the harmful-algal-bloom dinoflagellate Prorocentrum donghaiense. Marine Environmental Research 148, (2019) 19-25.
[47] J. Y. Zhu, H. Xiao, Q. Chen, M. Zhao, D. Sun, S. S. Duan, Growth Inhibition of Phaeocystis Globosa Induced by Luteolin-7-O-glucuronide from Seagrass Enhalus acoroides. International Journal of Environmental Research and Public Health 16, 14(2019).
[48] P. Chambonniere, J. Bronlund, B. Guieysse, Pathogen removal in high-rate algae pond: state of the art and opportunities. Journal of Applied Phycology 33, 3(2021) 1501-1511.
[49] X. Xiao, C. Li, H. M. Huang, Y. P. Lee, Inhibition effect of natural flavonoids on red tide alga Phaeocystis globosa and its quantitative structure-activity relationship. Environmental Science and Pollution Research 26, 23(2019) 23763-23776.
[50] S. Jeon, J. M. Lim, H. G. Lee, S. E. Shin, N. K. Kang, Y. I. Park, H. M. Oh, W. J. Jeong, B. R. Jeong, Y. K. Chang, Current status and perspectives of genome editing technology for microalgae. Biotechnology for Biofuels 10, (2017).
[51] L. M. Grattan, S. Holobaugh, J. G. Morris, Harmful algal blooms and public health. Harmful Algae 57, (2016) 2-8.
[52] P. V. L. Reddy, B. Kavitha, P. A. K. Reddy, K. H. Kim, TiO2-based photocatalytic disinfection of microbes in aqueous media: A review. Environmental Research 154, (2017) 296-303.
[53] L. Li, G. Pan, A Universal Method for Flocculating Harmful Algal Blooms in Marine and Fresh Waters Using Modified Sand. Environ. Sci. Technol. 47, 9(2013) 4555-4562.
[54] J. K. aEdzwald, Aluminum in Drinking Water: Occurrence, Effects, and Control. Journal American Water Works Association 112, 5(2020) 34-41.
[55] B. Bolto, J. Gregory, Organic polyelectrolytes in water treatment. Water Research 41, 11(2007) 2301-2324.
[56] J. Q. Jiang, The role of coagulation in water treatment. Current Opinion in Chemical Engineering 8, (2015) 36-44.
[57] M. Agbakpe, S. J. Ge, W. Zhang, X. Z. Zhang, P. Kobylarz, Algae harvesting for biofuel production: Influences of UV irradiation and polyethylenimine (PEI) coating on bacterial biocoagulation. Bioresource Technology 166, (2014) 266-272.
[58] S. B. Kurniawan, A. Ahmad, M. F. Imron, S. R. S. Abdullah, A. R. Othman, H. Abu Hasan, Potential of microalgae cultivation using nutrient-rich wastewater and harvesting performance by biocoagulants/bioflocculants: Mechanism, multi-conversion of biomass into valuable products, and future challenges. Journal of Cleaner Production 365, (2022).
Reviewer 2 Report
Jun Song et al reviewed importance of Nanoparticles, as an emerging control method for harmful algal blooms: Current technologies, challenges, and perspectives. This is very important and relevant topic as growing Harmful algal blooms (HABs) has become a global concern. The paper is drafted very well and ut can accepted after minor comments as suggested.
Comments:
· Line 35: please change ‘aquatic plant’ to ‘aquatic biota;
· Line 52: Please add each references for ultrasonic treatment, ultraviolet radiation, and membrane filtration.
· Line 62: please add references to ‘However, these…… some solution[…]
· Line 67-68: Please add each references for photocatalyst, flocculation and sedimentation, oxidation, Adsorption, nutrient recovery.
· Line 71: Please add each references for toxicity, bioaccumulation, and ecological impacts.
· From Chapter 2 until Chapter 3, no references in the all contents. Please add.
· I suggest to change caption for ‘scheme 1’ to ‘Figure 1’. And also place the diagram below information in Chapter 3.
· Line 276, Lanthanum is considered a promising rare-earth element owing to its abundant reserves and low price —> it is true? Please add the references
English is fine. Moderate changes are required.
Author Response
Reviewer #2:
Jun Song et al reviewed importance of Nanoparticles, as an emerging control method for harmful algal blooms: Current technologies, challenges, and perspectives. This is very important and relevant topic as growing Harmful algal blooms (HABs) has become a global concern. The paper is drafted very well and ut can accepted after minor comments as suggested.
Comments:
- Line 35: please change ‘aquatic plant’ to ‘aquatic biota;
Response 1:
Thanks for your careful review. We have referred to your valuable feedback and revised the wording in the manuscript.
- Line 52: Please add each references for ultrasonic treatment, ultraviolet radiation, and membrane filtration.
Response 2:
We do appreciate your valuable comments. We have added corresponding references in the newly submitted manuscript.
Reference
[7] P. Rajasekhar, L. H. Fan, T. Nguyen, F. A. Roddick, A review of the use of sonication to control cyanobacterial blooms. Water Research 46, 14(2012) 4319-4329.
[8] M. L. Li, D. Y. Chen, Y. Liu, C. Y. Chuang, F. Z. Kong, P. J. Harrison, X. S. Zhu, Y. L. Jiang, Exposure of engineered nanoparticles to Alexandrium tamarense (Dinophyceae): Healthy impacts of nanoparticles via toxin-producing dinoflagellate. Sci. Total Environ. 610, (2018) 356-366.
[9] Y. H. Park, S. Kim, H. S. Kim, C. Park, Y. E. Choi, Adsorption Strategy for Removal of Harmful Cyanobacterial Species Microcystis aeruginosa Using Chitosan Fiber. Sustainability 12, 11(2020).
- Line 62: please add references to ‘However, these…… some solution[…]
Response 3:
Thanks for your careful and valuable reviewer. We have added some references to establish the viewpoint in the new submission.
Reference
[12] R. Sun, P. F. Sun, J. H. Zhang, S. Esquivel-Elizondo, Y. H. Wu, Microorganisms-based methods for harmful algal blooms control: A review. Bioresource Technology 248, (2018) 12-20.
[13] H. Mohan, S. Vadivel, S. Rajendran, Removal of harmful algae in natural water by semiconductor photocatalysis- A critical review. Chemosphere 302, (2022).
- Line 67-68: Please add each references for photocatalyst, flocculation and sedimentation, oxidation, Adsorption, nutrient recovery.
Response 4:
Thank your so much for your valuable suggestions. We have added the cooresponding reference in the newly manuscript.
Reference
[15] F. A. Kibuye, A. Zamyadi, E. C. Wert, A critical review on operation and performance of source water control strategies for cyanobacterial blooms: Part I-chemical control methods. Harmful Algae 109, (2021).
[16] Q. Yue, X. W. He, N. Yan, S. D. Tian, C. C. Liu, W. X. Wang, L. Luo, B. Z. Tang, Photodynamic control of harmful algal blooms by an ultra-efficient and degradable AIEgen-based photosensitizer. Chemical Engineering Journal 417, (2021).
[17] B. X. Ren, K. A. Weitzel, X. D. Duan, M. N. Nadagouda, D. D. Dionysiou, A comprehensive review on algae removal and control by coagulation-based processes: mechanism, material, and application. Separation and Purification Technology 293, (2022).
[18] J. Suazo-Hernandez, P. Sepulveda, L. Caceres-Jensen, J. Castro-Rojas, P. Poblete-Grant, N. Bolan, M. D. Mora, nZVI-Based Nanomaterials Used for Phosphate Removal from Aquatic Systems. Nanomaterials 13, 3(2023).
- Line 71: Please add each references for toxicity, bioaccumulation, and ecological impacts.
Response 5:
We do appreciate your valuable comments. We have added corresponding references in the newly submitted manuscript.
Reference
[19] V. Aruoja, S. Pokhrel, M. Sihtmae, M. Mortimer, L. Madler, A. Kahru, Toxicity of 12 metal-based nanoparticles to algae, bacteria and protozoa. Environmental Science-Nano 2, 6(2015) 630-644.
[20] Y. X. Wang, X. S. Zhu, Y. M. Lao, X. H. Lv, Y. Tao, B. M. Huang, J. X. Wang, J. Zhou, Z. H. Cai, TiO2 nanoparticles in the marine environment: Physical effects responsible for the toxicity on algae Phaeodactylum tricornutum. Sci. Total Environ. 565, (2016) 818-826.
[21] R. J. Miller, E. B. Muller, B. Cole, T. Martin, R. Nisbet, G. K. Bielmyer-Fraser, T. A. Jarvis, A. A. Keller, G. Cherr, H. S. Lenihan, Photosynthetic efficiency predicts toxic effects of metal nanomaterials in phytoplankton. Aquatic Toxicology 183, (2017) 85-93.
[22] E. Priyadarshini, S. S. Priyadarshini, N. Pradhan, Heavy metal resistance in algae and its application for metal nanoparticle synthesis. Applied Microbiology and Biotechnology 103, 8(2019) 3297-3316.
- From Chapter 2 until Chapter 3, no references in the all contents. Please add.
Response 6:
Thanks for your meticulous review. We have supplemented the missing references in Chapters 2 and 3.
Reference
[25] H. W. Paerl, T. G. Otten, Harmful Cyanobacterial Blooms: Causes, Consequences, and Controls. Microbial Ecology 65, 4(2013) 995-1010.
[26] G. Hallegraeff, H. Enevoldsen, A. Zingone, Global harmful algal bloom status reporting. Harmful Algae 102, (2021) 3.
[27] H. W. Paerl, J. Huisman, Climate change: a catalyst for global expansion of harmful cyanobacterial blooms. Environmental Microbiology Reports 1, 1(2009) 27-37.
[28] W. W. Carmichael, Health effects of toxin-producing cyanobacteria: "The CyanoHABs". Human and Ecological Risk Assessment 7, 5(2001) 1393-1407.
[29] M. Pal, P. J. Yesankar, A. Dwivedi, A. Qureshi, Biotic control of harmful algal blooms (HABs): A brief review. Journal of Environmental Management 268, (2020).
[30] T. W. Davis, D. L. Berry, G. L. Boyer, C. J. Gobler, The effects of temperature and nutrients on the growth and dynamics of toxic and non-toxic strains of Microcystis during cyanobacteria blooms. Harmful Algae 8, 5(2009) 715-725.
[31] D. G. Schmale, A. P. Ault, W. Saad, D. T. Scott, J. A. Westrick, Perspectives on Harmful Algal Blooms (HABs) and the Cyberbiosecurity of Freshwater Systems. Frontiers in Bioengineering and Biotechnology 7, (2019).
[32] Y. Z. Tang, H. F. Gu, Z. H. Wang, D. Y. Liu, Y. Wang, D. D. Lu, Z. X. Hu, Y. Y. Deng, L. X. Shang, Y. Z. Qi, Exploration of resting cysts (stages) and their relevance for possibly HABs-causing species in China. Harmful Algae 107, (2021).
[33] N. Xu, M. Wang, Y. Z. Tang, Q. Zhang, S. S. Duan, C. J. Gobler, Acute toxicity of the cosmopolitan bloom-forming dinoflagellate Akashiwo sanguinea to finfish, shellfish, and zooplankton. Aquatic Microbial Ecology 80, 3(2018) 209-222.
[34] N. Inaba, V. L. Trainer, Y. Onishi, K. I. Ishii, S. Wyllie-Echeverria, I. Imai, Algicidal and growth-inhibiting bacteria associated with seagrass and macroalgae beds in Puget Sound, WA, USA. Harmful Algae 62, (2017) 136-147.
[35] C. R. C. Kouakou, T. G. Poder, Economic impact of harmful algal blooms on human health: a systematic review. Journal of Water and Health 17, 4(2019) 499-516.
[36] C. Belin, D. Soudant, Z. Amzil, Three decades of data on phytoplankton and phycotoxins on the French coast: Lessons from REPHY and REPHYTOX. Harmful Algae 102, (2021).
[37] A. Zingone, L. Escalera, K. Aligizaki, M. Fernandez-Tejedor, A. Ismael, M. Montresor, P. Mozetic, S. Tas, C. Totti, Toxic marine microalgae and noxious blooms in the Mediterranean Sea: A contribution to the Global HAB Status Report. Harmful Algae 102, (2021).
[38] S. Sakamoto, W. A. Lim, D. D. Lu, X. F. Dai, T. Orlova, M. Iwataki, Harmful algal blooms and associated fisheries damage in East Asia: Current status and trends in China, Japan, Korea and Russia. Harmful Algae 102, (2021).
[39] C. Wang, Z. Y. Wang, P. F. Wang, S. H. Zhang, Multiple Effects of Environmental Factors on Algal Growth and Nutrient Thresholds for Harmful Algal Blooms: Application of Response Surface Methodology. Environmental Modeling & Assessment 21, 2(2016) 247-259.
[40] I. Sunesen, S. M. Mendez, J. E. Mancera-Pineda, M. Y. D. Bottein, H. Enevoldsen, The Latin America and Caribbean HAB status report based on OBIS and HAEDAT maps and databases. Harmful Algae 102, (2021).
[41] T. C. Daniel, A. N. Sharpley, J. L. Lemunyon, Agricultural phosphorus and eutrophication: A symposium overview. Journal of Environmental Quality 27, 2(1998) 251-257.
[42] M. Rathod, K. Mody, S. Basha, Efficient removal of phosphate from aqueous solutions by red seaweed, Kappaphycus alverezii. Journal of Cleaner Production 84, (2014) 484-493.
[43] B. L. Wu, J. Wan, Y. Y. Zhang, B. C. Pan, I. M. C. Lo, Selective Phosphate Removal from Water and Wastewater using Sorption: Process Fundamentals and Removal Mechanisms. Environ. Sci. Technol. 54, 1(2020) 50-66.
[44] X. Y. Li, Y. H. Xie, F. Jiang, B. Wang, Q. L. Hu, Y. Tang, T. Luo, T. Wu, Enhanced phosphate removal from aqueous solution using resourceable nano-CaO2/BC composite: Behaviors and mechanisms. Sci. Total Environ. 709, (2020).
[45] P. Loganathan, S. Vigneswaran, J. Kandasamy, N. S. Bolan, Removal and Recovery of Phosphate From Water Using Sorption. Critical Reviews in Environmental Science and Technology 44, 8(2014) 847-907.
[46] W. J. Xu, J. T. Wang, L. J. Tan, X. Guo, Q. N. Xue, Variation in allelopathy of extracellular compounds produced by Cylindrotheca closterium against the harmful-algal-bloom dinoflagellate Prorocentrum donghaiense. Marine Environmental Research 148, (2019) 19-25.
[47] J. Y. Zhu, H. Xiao, Q. Chen, M. Zhao, D. Sun, S. S. Duan, Growth Inhibition of Phaeocystis Globosa Induced by Luteolin-7-O-glucuronide from Seagrass Enhalus acoroides. International Journal of Environmental Research and Public Health 16, 14(2019).
[48] P. Chambonniere, J. Bronlund, B. Guieysse, Pathogen removal in high-rate algae pond: state of the art and opportunities. Journal of Applied Phycology 33, 3(2021) 1501-1511.
[49] X. Xiao, C. Li, H. M. Huang, Y. P. Lee, Inhibition effect of natural flavonoids on red tide alga Phaeocystis globosa and its quantitative structure-activity relationship. Environmental Science and Pollution Research 26, 23(2019) 23763-23776.
[50] S. Jeon, J. M. Lim, H. G. Lee, S. E. Shin, N. K. Kang, Y. I. Park, H. M. Oh, W. J. Jeong, B. R. Jeong, Y. K. Chang, Current status and perspectives of genome editing technology for microalgae. Biotechnology for Biofuels 10, (2017).
[51] L. M. Grattan, S. Holobaugh, J. G. Morris, Harmful algal blooms and public health. Harmful Algae 57, (2016) 2-8.
[52] P. V. L. Reddy, B. Kavitha, P. A. K. Reddy, K. H. Kim, TiO2-based photocatalytic disinfection of microbes in aqueous media: A review. Environmental Research 154, (2017) 296-303.
[53] L. Li, G. Pan, A Universal Method for Flocculating Harmful Algal Blooms in Marine and Fresh Waters Using Modified Sand. Environ. Sci. Technol. 47, 9(2013) 4555-4562.
[54] J. K. aEdzwald, Aluminum in Drinking Water: Occurrence, Effects, and Control. Journal American Water Works Association 112, 5(2020) 34-41.
[55] B. Bolto, J. Gregory, Organic polyelectrolytes in water treatment. Water Research 41, 11(2007) 2301-2324.
[56] J. Q. Jiang, The role of coagulation in water treatment. Current Opinion in Chemical Engineering 8, (2015) 36-44.
[57] M. Agbakpe, S. J. Ge, W. Zhang, X. Z. Zhang, P. Kobylarz, Algae harvesting for biofuel production: Influences of UV irradiation and polyethylenimine (PEI) coating on bacterial biocoagulation. Bioresource Technology 166, (2014) 266-272.
[58] S. B. Kurniawan, A. Ahmad, M. F. Imron, S. R. S. Abdullah, A. R. Othman, H. Abu Hasan, Potential of microalgae cultivation using nutrient-rich wastewater and harvesting performance by biocoagulants/bioflocculants: Mechanism, multi-conversion of biomass into valuable products, and future challenges. Journal of Cleaner Production 365, (2022).
- I suggest to change caption for ‘scheme 1’ to ‘Figure 1’. And also place the diagram below information in Chapter 3.
Response 7:
Thanks for your careful review. We have taken into consideration your valuable suggestions and have made revisions to the placement of Figure 1 in the new manuscript.
- Line 276, Lanthanum is considered a promising rare-earth element owing to its abundant reserves and low price —> it is true? Please add the references;
Response 8:
Thank you for your valuable comments. We apologize if this may have caused any misunderstanding. In fact, what we meant is that lanthanum is inexpensive among rare earth elements, and its reserves are indeed abundant. To avoid any misconceptions, we have made certain modifications to our statement.
“[...] Due to its relatively low price and abundant reserves among rare-earth elements, lanthanum is considered a promising rare earth element. [...]”
Reference
[54] M. R. Razanajatovo, W. Y. Gao, Y. R. Song, X. Zhao, Q. N. Sun, Q. R. Zhang, Selective adsorption of phosphate in water using lanthanum-based nanomaterials: A critical review. Chinese Chemical Letters 32, 9(2021) 2637-2647.
Reviewer 3 Report
The present review article concerns the use of nanoparticles to treat harmful algal blooms. The topic of the article is important - algal blooms are encountered more and more often due to e.g. climate changes and/or human activity. Authors present a complex approach to the topic. They describe various aspects of the issue, starting from causes and effects of algal blooms, through possible treatments, to the use of nanoparticles. The article might be a good reference point for those seeking information on the problem.
Chapters 2 and 3 lack references to articles. They should be added to increase credibility of the manuscript.
Furthermore, some recent studies are not reviewed, e.g. Materials Today: Proceedings, 2023, https://doi.org/10.1016/j.matpr.2023.01.008. Please search literature database and consider to include the most recent items.
Some time ago, a review article covering similar area was published: Environ. Sci.: Nano, 2019,6, 1026-1042. Authors are advised to cite major publications that are reported therein. A citation to this article might also enrich conclusions section.
Author Response
Reviewer #3:
The present review article concerns the use of nanoparticles to treat harmful algal blooms. The topic of the article is important - algal blooms are encountered more and more often due to e.g. climate changes and/or human activity. Authors present a complex approach to the topic. They describe various aspects of the issue, starting from causes and effects of algal blooms, through possible treatments, to the use of nanoparticles. The article might be a good reference point for those seeking information on the problem.
Chapters 2 and 3 lack references to articles. They should be added to increase credibility of the manuscript.
Furthermore, some recent studies are not reviewed, e.g. Materials Today: Proceedings, 2023, https://doi.org/10.1016/j.matpr.2023.01.008. Please search literature database and consider to include the most recent items.
Some time ago, a review article covering similar area was published: Environ. Sci.: Nano, 2019,6, 1026-1042. Authors are advised to cite major publications that are reported therein. A citation to this article might also enrich conclusions section.
Response:
Thank you for your meticulous review. We have supplemented the missing references in Chapters 2 and 3. We greatly appreciate the valuable information you provided again. We have referenced the two articles you suggested and also incorporated some of the information cited within them.
Reference
[14] F. R. Chen, Z. G. Xiao, L. Yue, J. Wang, Y. Feng, X. S. Zhu, Z. Y. Wang, B. S. Xing, Algae response to engineered nanoparticles: current understanding, mechanisms and implications. Environmental Science-Nano 6, 4(2019) 1026-1042.
[24] A. H.M, M. N. A. Uda, S. C. B. Gopinath, Z. A. Arsat, F. Abdullah, M. F. A. Muttalib, M. K. R. Hashim, U. Hashim, M. N. A. Uda, A. R. W. Yaakub, N. H. Ibrahim, N. A. Parmin, T. Adam, Green route synthesis of antimicrobial nanoparticles using sewage alga bloom. Materials Today: Proceedings (2023).
[25] H. W. Paerl, T. G. Otten, Harmful Cyanobacterial Blooms: Causes, Consequences, and Controls. Microbial Ecology 65, 4(2013) 995-1010.
[26] G. Hallegraeff, H. Enevoldsen, A. Zingone, Global harmful algal bloom status reporting. Harmful Algae 102, (2021) 3.
[27] H. W. Paerl, J. Huisman, Climate change: a catalyst for global expansion of harmful cyanobacterial blooms. Environmental Microbiology Reports 1, 1(2009) 27-37.
[28] W. W. Carmichael, Health effects of toxin-producing cyanobacteria: "The CyanoHABs". Human and Ecological Risk Assessment 7, 5(2001) 1393-1407.
[29] M. Pal, P. J. Yesankar, A. Dwivedi, A. Qureshi, Biotic control of harmful algal blooms (HABs): A brief review. Journal of Environmental Management 268, (2020).
[30] T. W. Davis, D. L. Berry, G. L. Boyer, C. J. Gobler, The effects of temperature and nutrients on the growth and dynamics of toxic and non-toxic strains of Microcystis during cyanobacteria blooms. Harmful Algae 8, 5(2009) 715-725.
[31] D. G. Schmale, A. P. Ault, W. Saad, D. T. Scott, J. A. Westrick, Perspectives on Harmful Algal Blooms (HABs) and the Cyberbiosecurity of Freshwater Systems. Frontiers in Bioengineering and Biotechnology 7, (2019).
[32] Y. Z. Tang, H. F. Gu, Z. H. Wang, D. Y. Liu, Y. Wang, D. D. Lu, Z. X. Hu, Y. Y. Deng, L. X. Shang, Y. Z. Qi, Exploration of resting cysts (stages) and their relevance for possibly HABs-causing species in China. Harmful Algae 107, (2021).
[33] N. Xu, M. Wang, Y. Z. Tang, Q. Zhang, S. S. Duan, C. J. Gobler, Acute toxicity of the cosmopolitan bloom-forming dinoflagellate Akashiwo sanguinea to finfish, shellfish, and zooplankton. Aquatic Microbial Ecology 80, 3(2018) 209-222.
[34] N. Inaba, V. L. Trainer, Y. Onishi, K. I. Ishii, S. Wyllie-Echeverria, I. Imai, Algicidal and growth-inhibiting bacteria associated with seagrass and macroalgae beds in Puget Sound, WA, USA. Harmful Algae 62, (2017) 136-147.
[35] C. R. C. Kouakou, T. G. Poder, Economic impact of harmful algal blooms on human health: a systematic review. Journal of Water and Health 17, 4(2019) 499-516.
[36] C. Belin, D. Soudant, Z. Amzil, Three decades of data on phytoplankton and phycotoxins on the French coast: Lessons from REPHY and REPHYTOX. Harmful Algae 102, (2021).
[37] A. Zingone, L. Escalera, K. Aligizaki, M. Fernandez-Tejedor, A. Ismael, M. Montresor, P. Mozetic, S. Tas, C. Totti, Toxic marine microalgae and noxious blooms in the Mediterranean Sea: A contribution to the Global HAB Status Report. Harmful Algae 102, (2021).
[38] S. Sakamoto, W. A. Lim, D. D. Lu, X. F. Dai, T. Orlova, M. Iwataki, Harmful algal blooms and associated fisheries damage in East Asia: Current status and trends in China, Japan, Korea and Russia. Harmful Algae 102, (2021).
[39] C. Wang, Z. Y. Wang, P. F. Wang, S. H. Zhang, Multiple Effects of Environmental Factors on Algal Growth and Nutrient Thresholds for Harmful Algal Blooms: Application of Response Surface Methodology. Environmental Modeling & Assessment 21, 2(2016) 247-259.
[40] I. Sunesen, S. M. Mendez, J. E. Mancera-Pineda, M. Y. D. Bottein, H. Enevoldsen, The Latin America and Caribbean HAB status report based on OBIS and HAEDAT maps and databases. Harmful Algae 102, (2021).
[41] T. C. Daniel, A. N. Sharpley, J. L. Lemunyon, Agricultural phosphorus and eutrophication: A symposium overview. Journal of Environmental Quality 27, 2(1998) 251-257.
[42] M. Rathod, K. Mody, S. Basha, Efficient removal of phosphate from aqueous solutions by red seaweed, Kappaphycus alverezii. Journal of Cleaner Production 84, (2014) 484-493.
[43] B. L. Wu, J. Wan, Y. Y. Zhang, B. C. Pan, I. M. C. Lo, Selective Phosphate Removal from Water and Wastewater using Sorption: Process Fundamentals and Removal Mechanisms. Environ. Sci. Technol. 54, 1(2020) 50-66.
[44] X. Y. Li, Y. H. Xie, F. Jiang, B. Wang, Q. L. Hu, Y. Tang, T. Luo, T. Wu, Enhanced phosphate removal from aqueous solution using resourceable nano-CaO2/BC composite: Behaviors and mechanisms. Sci. Total Environ. 709, (2020).
[45] P. Loganathan, S. Vigneswaran, J. Kandasamy, N. S. Bolan, Removal and Recovery of Phosphate From Water Using Sorption. Critical Reviews in Environmental Science and Technology 44, 8(2014) 847-907.
[46] W. J. Xu, J. T. Wang, L. J. Tan, X. Guo, Q. N. Xue, Variation in allelopathy of extracellular compounds produced by Cylindrotheca closterium against the harmful-algal-bloom dinoflagellate Prorocentrum donghaiense. Marine Environmental Research 148, (2019) 19-25.
[47] J. Y. Zhu, H. Xiao, Q. Chen, M. Zhao, D. Sun, S. S. Duan, Growth Inhibition of Phaeocystis Globosa Induced by Luteolin-7-O-glucuronide from Seagrass Enhalus acoroides. International Journal of Environmental Research and Public Health 16, 14(2019).
[48] P. Chambonniere, J. Bronlund, B. Guieysse, Pathogen removal in high-rate algae pond: state of the art and opportunities. Journal of Applied Phycology 33, 3(2021) 1501-1511.
[49] X. Xiao, C. Li, H. M. Huang, Y. P. Lee, Inhibition effect of natural flavonoids on red tide alga Phaeocystis globosa and its quantitative structure-activity relationship. Environmental Science and Pollution Research 26, 23(2019) 23763-23776.
[50] S. Jeon, J. M. Lim, H. G. Lee, S. E. Shin, N. K. Kang, Y. I. Park, H. M. Oh, W. J. Jeong, B. R. Jeong, Y. K. Chang, Current status and perspectives of genome editing technology for microalgae. Biotechnology for Biofuels 10, (2017).
[51] L. M. Grattan, S. Holobaugh, J. G. Morris, Harmful algal blooms and public health. Harmful Algae 57, (2016) 2-8.
[52] P. V. L. Reddy, B. Kavitha, P. A. K. Reddy, K. H. Kim, TiO2-based photocatalytic disinfection of microbes in aqueous media: A review. Environmental Research 154, (2017) 296-303.
[53] L. Li, G. Pan, A Universal Method for Flocculating Harmful Algal Blooms in Marine and Fresh Waters Using Modified Sand. Environ. Sci. Technol. 47, 9(2013) 4555-4562.
[54] J. K. aEdzwald, Aluminum in Drinking Water: Occurrence, Effects, and Control. Journal American Water Works Association 112, 5(2020) 34-41.
[55] B. Bolto, J. Gregory, Organic polyelectrolytes in water treatment. Water Research 41, 11(2007) 2301-2324.
[56] J. Q. Jiang, The role of coagulation in water treatment. Current Opinion in Chemical Engineering 8, (2015) 36-44.
[57] M. Agbakpe, S. J. Ge, W. Zhang, X. Z. Zhang, P. Kobylarz, Algae harvesting for biofuel production: Influences of UV irradiation and polyethylenimine (PEI) coating on bacterial biocoagulation. Bioresource Technology 166, (2014) 266-272.
[58] S. B. Kurniawan, A. Ahmad, M. F. Imron, S. R. S. Abdullah, A. R. Othman, H. Abu Hasan, Potential of microalgae cultivation using nutrient-rich wastewater and harvesting performance by biocoagulants/bioflocculants: Mechanism, multi-conversion of biomass into valuable products, and future challenges. Journal of Cleaner Production 365, (2022).